# What Froze the Genetic Code?

**DOI:** 10.3390/life7020014

**Published:** 2017-04-05

**Authors:** Lluís Ribas de Pouplana, Adrian Gabriel Torres, Àlbert Rafels-Ybern

**Affiliations:** 1Institute for Research in Biomedicine (IRB Barcelona), The Barcelona Institute of Science and Technology, Baldiri Reixac, 10, 08028 Barcelona, Spain; adriangabriel.torres@irbbarcelona.org (A.G.T.); albert.rafels@irbbarcelona.org (À.R.-Y.); 2Catalan Institution for Research and Advanced Studies (ICREA), Passeig Lluis Companys 23, 08010 Barcelona, Spain

**Keywords:** translation, evolution, speciation, protein folds, tRNA, ribosome

## Abstract

The frozen accident theory of the Genetic Code was a proposal by Francis Crick that attempted to explain the universal nature of the Genetic Code and the fact that it only contains information for twenty amino acids. Fifty years later, it is clear that variations to the universal Genetic Code exist in nature and that translation is not limited to twenty amino acids. However, given the astonishing diversity of life on earth, and the extended evolutionary time that has taken place since the emergence of the extant Genetic Code, the idea that the translation apparatus is for the most part immobile remains true. Here, we will offer a potential explanation to the reason why the code has remained mostly stable for over three billion years, and discuss some of the mechanisms that allow species to overcome the intrinsic functional limitations of the protein synthesis machinery.

## 1. The Limits of the Genetic Code

The race to identify the structure of the Genetic Code was intense. However, the literature of the time suggests that it was, nevertheless, a collaborative exercise enriched by an intense academic debate that tried to offer explanations to the many questions that kept popping up. 

In his seminal paper ‘The origin of the Genetic Code’, Francis Harry Compton Crick offered a good example of this dynamic as he and Leslie Orgel published their respective views on this topic in back-to-back papers [1,2]. In his paper Crick used the term ‘frozen accident’ to refer to the apparent inability of the code to accept new variations, and he contrasted this hypothesis with an alternative possibility: the stereochemical theory for the origin of the Genetic Code.

In the forty-nine years that have passed since the publication of this paper, we have advanced very significantly in our understanding of the molecular mechanisms that govern the Genetic Code. However, many fundamental questions regarding the origin and evolution of the code remain open, and chief among them is the reason why the system stopped incorporating new amino acids despite the obvious availability of codon sequences. 

Nevertheless, progress has been made. The remarkable advances in the structural analysis of ribosomes, tRNAs, and aminoacyl-tRNA synthetases (ARS) have led to several important conclusions regarding the central roles of RNA in the early Genetic Code, which persist today in the functions of transfer RNAs and the ribosome, among others [3,4,5]. We now have strong support for the notion that extant proteomes functionally replaced a preceding RNA world where most, if not all, biological catalysis was performed by RNA molecules [6].

It is generally accepted that a primitive Genetic Code, using a limited number of amino acids or groups of related amino acids under a single identity, expanded through the generation of new tRNA identities that increased the number of residues being used, while allowing for a better discrimination between similar amino acid sidechains [7]. The remarkable clustering of chemically-related amino acids that can be seen in the Genetic Code possibly reflects the process of establishment of the different codon and tRNA identities, and is the basis for the coevolution theory of Wong [8,9]. 

It is reasonable to expect that the expansion of tRNA identities was accompanied by the evolution of tRNA-associated polypeptides (ancestors of extant ARS). Indeed, both the distribution of amino acids in the Genetic Code, as well as the structural features of tRNA, are closely mirrored by the organization of the two ARS classes [10]. 

It is possible that the initial interaction between primitive tRNAs and the ancestral forms of ARS was in a complex of tRNA molecules bound by a heterodimer from which the two families of ARS later would emerge [11]. It has also been proposed that these two ancestral ARS domains could be coded by complementary strands and, as such, be under tightly coupled selection [12]. This hypothesis can explain the broad internal organization of the two ARS classes, the intriguing distribution of amino acid specificities that can be seen within these same classes, and the many unexplained similarities in identity elements found between tRNAs that are aminoacylated by ARS of different classes [13,14].

## 2. Why Did the Genetic Code Freeze?

Given the extraordinary chemical diversity of biological amino acids, and the potential for a three-base code based on four bases to theoretically incorporate up to sixty-three amino acids, it is a priori unclear why the universal Genetic Code includes only twenty amino acids. This is even more puzzling if one considers that several additional amino acids, such as selenocysteine and pyrrolysine, are used for protein synthesis. Chemical modifications of side chains are widespread, suggesting that cells could use a larger repertoire of residues within the canonical Genetic Code. Thus, what drove the arrest in the emergence of new tRNA identities and the expansion of the Genetic Code?

Although faithful amino acid recognition is an essential feature of the Genetic Code, it is unlikely that it was a limiting factor in the growth of the system because the recognition is limited to the interactions with ARS active sites, which are extremely adaptable and supported by editing domains that can discriminate between similar side chains [15]. On the other hand, the recognition of tRNAs is a much harder challenge because the three-dimensional structures of all tRNAs are very similar, their chemical composition before modifications is more uniform, and the number of required specific interactions with protein components of the translation apparatus is much larger. 

We have proposed that a functional boundary exists with regards to the ability of the translation apparatus to successfully discriminate different tRNA identities. This boundary is determined by the overall capacity of the tRNA structure to incorporate different recognition elements. The incorporation of a new amino acid (hence a new tRNA identity) greatly increases the combinatorial problem faced by the translation machinery to specifically recognize individual tRNAs. This problem applies to modification enzymes, transport systems, ARS, elongation factors, ribosomes, etc. All tRNA identity elements need to coexist in a short RNA sequence whose structure is necessarily similar among all tRNAs in the cell. Additional constraints on tRNA evolution emerging from its non-canonical functions can also be envisaged. Our proposal is that this complex recognition network reaches a limit beyond which the incorporation of new tRNA identities is impossible without generating a recognition conflict with a pre-existing tRNA [16]. 

We have demonstrated that the saturation of structural and identity signals in a tRNA can prevent this molecule from incorporating other identities in evolution. We investigated the reasons for the intriguing lack of tRNA^Gly^_ACC_ in eukaryotic genomes and showed that pre-existing features of the tRNA^Gly^ anticodon loop are incompatible with the presence of an adenosine at position 34, explaining why an A34-containing tRNA could not evolve and become enriched in eukaryotes [16]. 

At the genomic level, we observed that species with low numbers of tRNA genes have significantly more nucleotide differences between their orthologous tRNA pairs than closely related species with a larger number of tRNA genes. This is consistent with the notion that an increase in complexity of tRNA populations leads to a higher conservation of tRNA sequences. Conversely, it would be expected that tRNA sequences would evolve faster in genomes with smaller numbers of tRNA genes. This situation is evident, for example, in mitochondria whose genomes have low numbers of tRNA genes [17]. Mitochondrial genomes display abundant deviations from the canonical Genetic Code, and contain the highest known variability in the structure and identity elements of tRNAs [18,19,20].

## 3. Evolutionary Strategies to Expand the Functional Boundaries of the Translation Apparatus

The study of globular protein folds has shown that the extant universe of proteins covers a minimal area of the vast potential number of protein structures. It is likely that extant protein structures evolved from the repetition of simpler domains that were assembled gradually through mechanisms of genetic recombination [21,22]. 

The synthesis of proteins generated through multiple repetitions of simple sequences may encounter difficulties due to the physicochemical characteristics of such repetitive peptides, or to the inability of tRNAs to maintain fidelity and reading frame when low complexity mRNA sequences are encountered. A number of adaptations have emerged to overcome some of these limitations. For example, the structure of the mammalian mitochondrial ribosome reveals that its polypeptide exit channel has been remodeled to allow not only the synthesis of the hydrophobic proteins that constitute the mitochondrial respiratory chain, but also their insertion into the mitochondrial membrane [23]. Also, EF-P (or eIF5A in eukaryotes), is a universally distributed elongation factor required for the translation of stretches of poly-proline codons [24]. Finally, translation of the extremely codon-biased mRNA transcripts coding for sericin and fibroin (protein components of silk) in the salivary glands of some arthropods requires a unique and highly skewed pool of cellular tRNAs, specifically selected to favor the translation of these mRNAs [25,26]. Thus, certain sequence combinations are a priori inaccessible to the translation apparatus, and functional improvements are needed to translate them.

The existence of species-specific adaptations of the translation apparatus indicate that some species have access to protein structures that are inaccessible to others [27]. We envisage that adaptations of the protein synthesis apparatus that allowed a given species to assemble new types of proteins would provide these organisms with the opportunity to evolve novel and unique functions (in the aforementioned example, the production of silk resulted in a novel mechanism for the growth and development of certain arthropods). We believe that this evolutionary process could start with a simple modification of the translation apparatus, which would allow species to increase their proteome diversity and drive speciation in a punctuated manner (Figure 1A).

Two important parameters that differentiate the translation apparatuses of the three domains of life are their genomic composition of tRNA genes and the set of base modifications in their mature tRNAs [28,29]. We have shown that the divergence of eukaryotic and bacterial genomes in terms of tRNA composition is tightly linked to the evolution of different base modifications in the two kingdoms [30]. In eukaryotic genomes, a remarkable enrichment in genes coding for A34-containing tRNA isoacceptors coincided with the appearance of heterodimeric adenosine deaminases acting on tRNAs (ADAT). This enzyme deaminates A34 to inosine (I34) in tRNAs decoding for eight different amino acids [31]. The activity of this enzyme allows the tRNA pool in eukaryotic cells to match the codon composition of their genomes [30]. 

In the human transcriptome, codons recognized by ADAT-modified tRNAs are significantly more abundant than those that do not require these modified tRNAs, and this preference is greater in proteins that are highly enriched in the eight amino acids that can be decoded by ADAT-modified tRNAs. We have shown that, in the human proteome, the polypeptides that display the highest preference for these ADAT-modified tRNAs contain extremely biased stretches of the amino acids threonine, alanine, proline, and serine (TAPS) [32]. Figure 1B shows an example of such proteins, Syndecan 3 (SDC3), a member of a proteoglycan family unique to placentals (Figure 1C). This observation suggests that the emergence of TAPS-enriched proteins in eukaryotes was facilitated by the evolutionary emergence of ADAT, which caused an ‘upgrade’ of the translation machinery through the modification of the composition and the codon-pairing capacity of their tRNA pool. We propose that the capacity of bacterial- and archaeal-type translation machineries to synthesize polypeptides highly enriched in TAPS, is limited by the functional characteristics of their tRNA pools, which may be either inefficient during the elongation phase of these transcripts (causing ribosomal stalling), or prone to decoding errors in these circumstances (causing deleterious levels of mutations in the resulting polypeptides).

The number of known species-specific features of the translation apparatus continues to grow and already includes the composition and regulation of several tRNA modifications, alterations to the ribosomal structure, the differential functionality of translation factors, and the protein and RNA composition of ribosomes, among others. Some of these adaptations may have resulted in translation machinery upgrades that allowed the synthesis of proteins with novel structures and functionality. The extent to which each of these differential features contributed to the divergence of proteomes is still unknown. However, a comparative analysis of the regions of the protein universe that are occupied by the proteomes of archaeal, bacterial, and eukaryotic organisms could shed light on this question.

In conclusion, the frozen accident that Francis Crick proposed with his characteristic genius may have been the result of the intrinsic limitations imposed by tRNA recognition, but translation has learned to overcome some of these initial limitations through additional functional adaptations that allow species to increase the range and roles of their proteins.

## Figures and Tables

**Figure 1 life-07-00014-f001:**
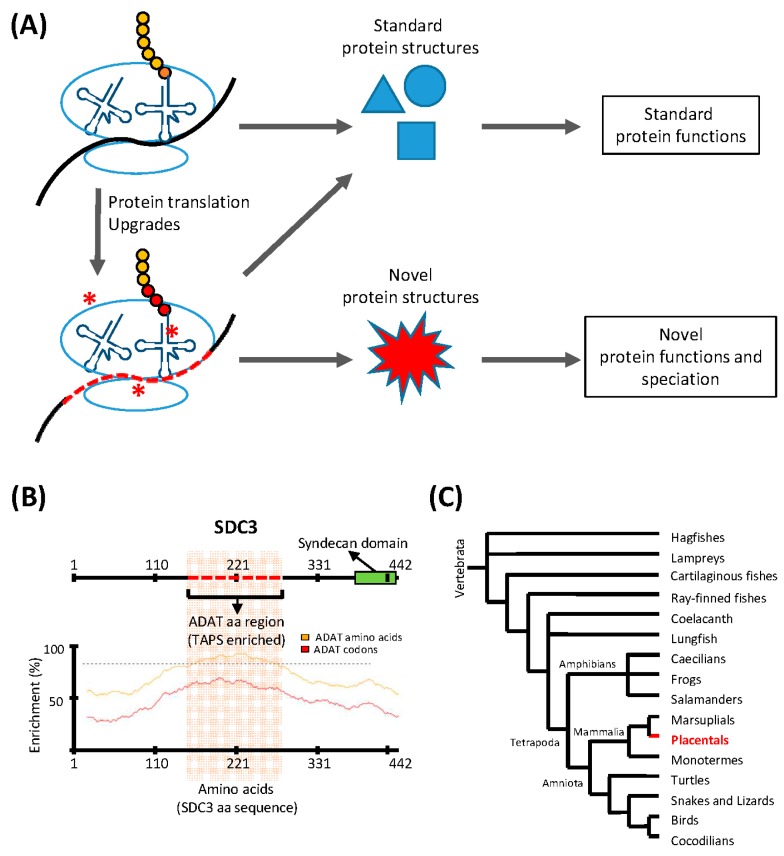
Translation upgrades may lead to novel protein structures and drive speciation. (**A**) The translation machinery is capable of synthesizing a finite number of standard protein structures, and translation ‘upgrades’ (red asterisks) such as codon usage adaptations, or modulation of the tRNA pool, allow the translation machinery to synthesize proteins with novel structures and functions. This process may drive speciation; (**B**) An example of a gene (SDC3) containing a region with a sequence highly enriched in ADAT-related amino acids (red dashed line; upper panel). The codon composition of the DNA coding for this domain is highly biased towards triplets recognized by tRNAs modified by ADAT. The lower panel shows the enrichment in ADAT-related amino acids (yellow line) and ADAT-dependent codons (red line) across the whole sequence of SDC3. The dashed line marks an enrichment level of ADAT-dependent codons of 80%; (**C**) Consensus phylogeny for Vertebrata. SDC3 belongs to the syndecan proteoglycan family found solely in placentals (highlighted region). The activity of ADAT may have contributed to the emergence of SDC3-type domains in placentals.

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
