# Peer review of "What Froze the Genetic Code?"

_life, 2017, doi:10.3390/life7020014_

Round 1

Reviewer 1 Report

Manuscript: What froze the genetic code?

Pouplana, Torres and Rafels-Ybern

Overall I found this to be a well articulated concept paper where the authors describe the concept and theoretic basis of the ‘frozen’ genetic code. While not an entirely unique concept, the authors do an exceptional job in utilizing the literature to address and support the key concepts and their opinions regarding tRNA evolution.

While I genuinely found this to be a very interesting read, I do have a couple points that could be addressed to make this a more complete manuscript.

While the main focus of the manuscript is on the limitations on tRNA diversity during evolution, the evolutionary constraints on these molecules that is set by the translational machinery is not fully developed. While some of this has been discussed in some detail in reference 15, it would be ideal in this manuscript to explicitly discuss some of these constraints in detail.

The manuscript does a great job in the describing some of the specialized translational machinery that has evolved for specific functions, but I am surprised there was not even a minor referral to “off-ribosome’ specialized functions of some tRNAs that have also evolved, which may also have had impact of tRNA evolution.

The authors can address these minor points in their manuscript if they believe it would add to their manuscript.

Author Response

We thank the reviewer for his support of the manuscript. He raises two very good points. Indeed, functional constraints must have played a major role in the evolution of tRNAs and the genetic code, and very possibly some of these constraints may be related to the many moonlighting roles of tRNA. Since this is an opinion piece we did not want to get into the details of describing this topic in depth, however we have added a phrase to the corresponding section stating exactly this point.

Reviewer 2 Report

The manuscript by Ribas de Pouplana et al, represents a very interesting point-of-view article regarding the evolution of the contemporary genetic code. I found the manuscript very well written and presented. The authors discuss in a comprehensive and concise way, several genomic and biochemical evidence to support their assumptions. At the same time they intrigue the reader with concepts that could be further explored experimentally. In addition, and given the manuscript length limitations, they provide adequate potential explanations for the limited divergence of the code, based on known mechanisms that allow species to enrich their proteomes with structural and functional diversifications. I feel that the manuscript is of potential interest to a broader readership and merits publication in its present form.

Author Response

We thank the reviewer for his/her support of the manuscript.

Reviewer 3 Report

In their concept paper, Ribas de Pouplana and coworkers give an overview of the possible reasons that could explain why the genetic code has remained so stable since its emergence and provide a potential explanation as to why this code, in most species, only encodes for the limited subset of 20 amino acids (aa) found in the standard/universal genetic code. They describe an example in which the synthesis of polypeptides that contain extremely biased repeats of threonine, alanine, proline and serine (TAPS) was overcome by the acquisition of a tRNA-modifying enzyme, ADAT, that deaminates A34 into I34. Acquisition of ADAT allowed enrichment of A34-containing tRNA isoacceptor genes (that became I34-containing tRNAs) in eukaryotes to match the codon composition of the genes that are enriched in the eight aa that can be decoded by I34-containing tRNAs among which TAPS. They propose that the difficulty of the translation machinery to translate repeats of TAPS codons was the evolutionary force that drove the emergence of the tRNA modifying enzyme ADAT that enhanced the codon-pairing capacity of the eukaryotes’ tRNA pool.

The manuscript is well-written and presents a nicely-articulated retrospective of the theories/data that could possibly explain why the genetic code remains so stable and only encodes a limited number of the cell’s aa repertoire. Their proposition that, in eukaryotes, emergence of ADAT reshaped the eukaryotes’ tRNA pool allowing translation of TAPS-rich polypeptides is convincing. I therefore recommend publication of the manuscript by Ribas de Pouplana and coworkers into life. I only have minor comments/theoretical questions for the authors.

Comments:

Page 1 line 41: I do agree with the authors that there is a consensus view that a preexisting RNA world in which primitive protein synthesis catalyzed by RNA molecules was replaced by a Ribonucleoprotein and protein world in which RNA has lost its central role in catalysis. In addition to reference #6, the authors could also refer to Suga’s pioneering work showing that aminoacylation can be catalyzed by a ribozyme (flexizyme) which nicely illustrates the part about the preexisting RNA world.

Page 2 paragraph beginning with line 55: One of the hypothesis for the perfect partition into 10 class 1 and 10 class 2 aaRSs is that in a single gene, one of DNA strand would have encoded an aaRS of a given class classes while the other would have encoded the aaRS of the other class. A succession of duplications of this gene would have then led to 10 class 1 and class 2 aaRSs.  The group of C.W. Carter reported a nice example that illustrates this hypothesis of a sense-antisense ancestry and that the authors should quote (Pham et al 2007, Mol. Cell).

Page 2 paragraph beginning with line 77: In their paper (ref # 55) Saint-Léger and coworkers argue that of the 76 bases of a tRNA 23 are invariant mainly because they are required for maintenance of the L-shape of the tRNA recognition by translation factors and by the ribosome itself, leaving 53 bases that can be combined to delineate the required tRNA identity sets for each of the aaRSs. They also propose that from these 53 remaining bases that could be used to set a tRNA identity for a given aaRS, a yet undetermined number of bases cannot be used as tRNA identity determinants because they are used as anti-determinants, posttranscriptional modification transport or even functions that have still to be unraveled. While I do agree with most of these assumptions/data, synthesis of an aminoacyl-tRNA does not solely proceed through a one-step attachment of an amino acid onto its corresponding tRNA by the cognate aaRS. 4 out the 20 genetically encoded aa (Asn, Gln, Cys and Sec) are charged onto their cognate tRNAs through two-steps pathways that involve mischarging of a precursor aa onto the tRNA and then tRNA-dependent modification of the precursor aa into the cognate one by a second enzyme. Proceeding through a two-steps aminoacylation route in which the aa chemical groups would be part of the identity elements enriches the chemical diversity that can overcome the limitations of the tRNA bases that can be used for evolving a new tRNA identity. This strategy has also the advantage to enable the use of a codon that has already be assigned to encode the new aa rather than the use of a stop codon. This was, for example, the case for the reassignment of the primitive AAU/C Asp codons and CAA/G Glu codons to respectively Asn and Gln. These two-steps routes through the control of the efficiency of the tRNA-dependent modifying enzyme enable modulation of the ambiguity of the codon that will be reassigned in order not to degenerate the proteome of the organism to an extend that would be lethal otherwise, until the time at which the reassignment can be fixed. I would like the authors' opinion on this possibility.

Author Response

We thank the reviewer for his/her comments on our paper and his support. Regarding the specific points raised:

Mention the flexizyme. We are great admirers of the Suga work, and take advantage of the Flexizyme often. We also understand that the Flexizyme offers a good example of how extant tRNA introns could have evolved from preceding RNA catalytic domains. However, since there are many other examples of selected catalytic RNAs that also deserve recognition, we would rather just cite a review of the field that can help readers find good examples of support for the RNA world theory.

Mention ARS urenzymes. We agree that this is an important discovery regarding ARS evolution, and have added a phrase regarding this topic, and the reference, to the text.

 Gln and Asn as examples of tRNA limitations. We agree with the reviewer, the parallel mechanisms required for the aminoacylation of tRNAs with amino acids Asn, Gln, Cys and Sec are likely strategies to overcome saturation of identity elements , and prevent the accumulation of proteome errors due to mischarging.

Round 2

Reviewer 1 Report

I am satisfied with the current version of the manuscript.

Reviewer 3 Report

As the authors satisfactorily answered to the questions and issues raised upon initial reviewing, the revised version of the manuscript by Ribas de Pouplana and coworkers is now suitable for publication into life.